# Provoking Artificial Slips and Trips towards Perturbation-Based Balance Training: A Narrative Review

**DOI:** 10.3390/s22239254

**Published:** 2022-11-28

**Authors:** Rafael N. Ferreira, Nuno Ferrete Ribeiro, Joana Figueiredo, Cristina P. Santos

**Affiliations:** 1Center for MicroElectroMechanical Systems, University of Minho, 4800-058 Guimarães, Portugal; 2LABBELS—Associate Laboratory, 4710-057 Braga, Portugal; 3LABBELS—Associate Laboratory, 4800-058 Guimarães, Portugal; 4MIT Portugal Program, School of Engineering, University of Minho, 4800-058 Guimarães, Portugal

**Keywords:** provoked falls, perturbation-based balance training, slips, trips, fall prevention

## Abstract

Humans’ balance recovery responses to gait perturbations are negatively impacted with ageing. Slip and trip events, the main causes preceding falls during walking, are likely to produce severe injuries in older adults. While traditional exercise-based interventions produce inconsistent results in reducing patients’ fall rates, perturbation-based balance training (PBT) emerges as a promising task-specific solution towards fall prevention. PBT improves patients’ reactive stability and fall-resisting skills through the delivery of unexpected balance perturbations. The adopted perturbation conditions play an important role towards PBT’s effectiveness and the acquisition of meaningful sensor data for studying human biomechanical reactions to loss of balance (LOB) events. Hence, this narrative review aims to survey the different methods employed in the scientific literature to provoke artificial slips and trips in healthy adults during treadmill and overground walking. For each type of perturbation, a comprehensive analysis was conducted to identify trends regarding the most adopted perturbation methods, gait phase perturbed, gait speed, perturbed leg, and sensor systems used for data collection. The reliable application of artificial perturbations to mimic real-life LOB events may reduce the gap between laboratory and real-life falls and potentially lead to fall-rate reduction among the elderly community.

## 1. Introduction

The second main cause of unintentional injury-based deaths worldwide is related to fall events [1]. Each year, 37-million people experience at least one fall. Falls may compromise an individual’s physical condition and lead to fear of falling, resulting in physical inactivity and social isolation [2]. The elderly age group entails the highest fall risk associated with cognitive, physical, and sensory deficits, which arise with ageing [1]. Thereby, the unpredictability of fall risk events that can occur across a wide range of scenarios during everyday living presents a persistent threat to the quality of life of older individuals. There exists a need to prepare high fall-risk individuals to successfully overcome unexpected gait perturbations since walking is the most common activity preceding fall-related events [3,4,5].

Previous literature studies have highlighted that exercise-based interventions may help in fall prevention. However, they produce inconsistent results in reducing patients’ fall rates due to their lack of specificity to target specific neuromuscular mechanisms underlying balance recovery reactions following loss of balance (LOB) events [3,6,7]. Hence, perturbation-based balance training (PBT) arises as a promising task-specific solution. It comprises the delivery of unexpected destabilising balance perturbations in a controlled environment simulating real-life LOB scenarios, which improves subjects’ reactive stability and fall-resisting skills [7,8]. In a previous review study [7], the authors compared several methods regarding the promotion of balance in the elderly population. Based on the evidence found, it was suggested that recent approaches such as PBT may be more effective in promoting balance recovery reactions after perturbations than traditional methods such as resistance training. Further literature studies have shown the effectiveness of the PBT paradigm on producing immediate [9,10] and long-term [11,12] balance and stability improvements as well as fall-rate reduction. The observed growth of PBT throughout recent years is depicted by the increasing amount of studies that consider this type of rehabilitation during their experiments. PBT has been used to improve balance recovery in participants with various diseases, such as stroke [13], spinal cord injury [14], Parkinson’s [15], cerebral palsy [16], back pain [17], and obesity [18].

PBT must include the application of perturbations that resemble the most relevant causes of fall events in the real-life context. Slips and trips prevail as the most common events that precede falls [5,19]. In this respect, PBT studies often rely on the application of these two main kinds of gait perturbations to the participants. Slips occur when the interface between the subject’s foot and the floor does not provide sufficient coefficient of friction (COF) [20]. These events take place mostly when the foot is either contacting or leaving the floor, which resemble critical body-weight transfer situations between the lower limbs, especially when the heel strikes the floor [20]. Therefore, literature studies provoke slip perturbations generally at the time of heel strike of the foot being perturbed [21,22]. Trip events occur when the motion of the swing limb is abruptly interrupted, which can be generally induced by encountering objects while walking [23]. Recent literature has focused more on addressing slip-related events rather than trips. In fact, slips have been identified as the main contributors to falls with a higher incidence than trips [24,25]. Previous investigations reported that slips accounted for 55% of falls on level surfaces, while trips only represented 22% [25].

The perturbation characteristics and the conditions in which they are applied play a fundamental role not only towards the effectiveness of PBT, concerning the improvement of balance recovery skills, but also to acquire meaningful data to better understand human biomechanical reactions to LOB events. A previous review conducted by McCrum et al. [3] studied the different gait perturbation methods used regarding PBT to improve reactive balance recovery in healthy older adults and reduce their fall rates. However, the authors applied narrow inclusion criteria regarding the ages of participants (mean age of at least 60 years), leaving only eight studies for the analysis. Therefore, review [3] does not analyse the methods used to provoke perturbations from studies that enrolled younger adult participants. Additionally, Karamanidis et al. [23] performed a review study that focused on balance training following slip- and trip-like PBT during treadmill locomotion. Despite including both slips and trips, information about how these LOB events are provoked during overground walking was not provided. Furthermore, in a recent review study conducted by McCrum et al. [26], key concepts and common concerns related to PBT were discussed towards the implementation of this therapy in clinical settings. Despite the authors synthetically portraying the current evidence on the efficacy and feasibility of PBT, a deep analysis on the current systems and methodologies used to apply slips and trips was not performed.

The objective of this narrative review is to survey the different methods employed in the scientific literature to provoke slip- and trip-like perturbations in healthy adults during treadmill and overground walking and identify the key experimental aspects to consider in future related research. Hence, the following research questions were addressed: (i) “Which methods and walking conditions are used to provoke slip- and trip-like perturbations?”; (ii) “Is it preferable to deliver perturbations during treadmill or overground walking in PBT?”; (iii) “Is it preferable to use a single-belt or a split-belt treadmill to perturb walking?”; (iv) “What procedures are implemented to maintain responses to perturbations unbiased?”; (v) “Which limb is generally used to apply the perturbations?”; (vi) “Which was the participants’ walking speed during the trials?”; (vii) “What are the main sensor systems used to collect data during perturbation-based protocols?”; (viii) “Are there benefits to apply both slip- and trip-like perturbations?”; and (ix) What is the current feasibility of PBT implementation in clinical settings? This narrative review will provide novel literature analysis towards artificial slip and trip perturbations concerning the third, fourth, seventh, and eighth research questions since they were not addressed by McCrum et al. [3,26] and Karamanidis et al. [23]. Although the first and sixth questions have been analysed in review studies [3,23] and McCrum et al. [3] have also considered the fifth question, the work from this manuscript represents an attempt to identify whether new trends have been adopted in recent literature studies towards these investigation topics. Although review [26] analysed the second question, our review provides a deeper analysis on the comparison between perturbation delivery during treadmill and overground walking. Furthermore, although McCrum et al. [26] focused their analysis mainly on the ninth question, we attempted to conduct a more concise overview concerning the current stage of PBT implementation in clinical settings.

Overall, this narrative review will contribute with the latest knowledge on the best conditions to induce artificial slip and trip perturbations that may support future studies on PBT.

The remainder of this study is organised as follows: Section 2 describes the search strategy employed. Section 3 describes the search results regarding the methods found to provoke slip- and trip-like perturbations and also provides a brief overview of each perturbation method. Section 4 presents a general discussion of the search outcomes. Lastly, Section 5 provides the general conclusions obtained from this review.

## 2. Methods

A comprehensive search of the scientific literature was performed using Pubmed, Web of Science, CINAHL (EBSCO), and Scopus databases. This search was carried out until January 15 2021 using the set of keywords: (“gait OR walking OR walk OR locomotion”) AND (“perturb* OR trip* OR slip* OR balance loss OR dynamic stability OR static stability OR waist pull OR provoked falls”) AND (“training OR exercise OR adaptation OR adaptive OR repeated OR repetition OR rehabilitation OR task OR responses OR adjustments”) AND (“age OR ageing OR aging OR aged OR elderly OR old OR older OR senior”). The keywords used to perform this search were based on those used in a previous systematic review [3]. Since in that review paper, the search process was performed at the end of 2015, the present search considered all the articles that were published since 2016 to find updated trends or evidence regarding gait perturbation paradigms.

A total of 3622 articles were gathered from the aforementioned databases and 2288 remained after removal of duplicates. Afterwards, the papers obtained were screened based on their title. This process enabled the inclusion of articles that met the following inclusion criteria: (i) perturbations were applied exclusively during walking; (ii) perturbations were not visual nor was a virtual environment included in the experimental setup; and (iii) the paper was not a review. Reviews were excluded from the search results since the search strategy’s purpose was to find studies which described an experimental protocol for slip or trip perturbation delivery. Only studies that included healthy participants were included in order to enable more reliable comparisons across all studies collected during the search. Furthermore, the participants’ ages were not used as an exclusion criterion to achieve a comprehensive analysis of a wider range of methods used to deliver slip- and trip-like perturbations. A total of 338 articles resulted from the screening procedure. The article titles in which it was not clear whether the conditions stated above were respected were included in the abstract screening. The subsequent selection was based on the careful reading of the abstract of each remaining paper. The eligibility criteria were applied in order to obtain a set of articles that: (i) included only slip- and/or trip-like perturbations in the experimental protocol; (ii) delivered the perturbations unexpectedly; (iii) only included healthy subjects; and (iv) did not use an assistive robotic device during the experimental trials. Beyond these conditions, the criteria used for title screening were also applied during the abstract reading stage. A group of 110 articles was then obtained through the screening procedure. Since it is not possible to ascertain whether the papers fulfilled the eligibility criteria previously defined by only reading the title and abstract, the remaining articles were read and carefully analysed in order to exclude those that failed to meet at least one of the above mentioned conditions. After this analysis, a final group of 48 articles was obtained. Figure 1 depicts the Preferred Reporting Items for Systematic Review and Meta-Analysis (PRISMA) flowchart concerning the previously described literature search.

## 3. Results

From the 48 included studies, slip-like perturbations (40 studies) were more prevalent than trip-like perturbations (15 studies), and 7 studies performed both slip- and trip-like perturbations. We conducted a separate analysis for slip- and trip-like perturbations since they have different characteristics and their critical adverse effects are associated to different phases of gait [20,23,27,28]. Table 1 depicts the different methods used in the literature to deliver slip- and trip-like perturbations.

### 3.1. Slip-like Perturbations

Slip-like perturbations were issued in 40 studies, 18 and 29 manuscripts during treadmill and overground walking, respectively. Of these 40 manuscripts, 7 performed slip-like perturbations during both treadmill and overground locomotion. Figure 2 depicts some of the methods used to provoke slips in the selected studies.

#### 3.1.1. Treadmill Walking

Table 2 depicts the 18 studies that conducted slip-like perturbations during treadmill locomotion. Of these, 10 conducted their experiments using only young subjects (aged 40 or younger) [10,21,32,33,34,36,37,38,41,42], whereas 6 manuscripts only considered the enrolment of older participants (aged 60 or older) [9,11,30,31,35,40]. Additionally, two studies considered data from both young and older subjects [29,39]. Hereafter, if a study enrolled participants from different age groups, these groups were distinguished within the "Participants" column in the tables. Only 2 studies [9,11] enrolled more than 50 participants during the experimental trials. The sample size ranged from 10 to 152 participants with a median of approximately 28 subjects.

##### Perturbation Methods

All 18 treadmill studies provoked slips by changing the acceleration of the treadmill belt. This change focuses on the sudden decrease in belt velocity. In addition, some studies even reversed the direction of belt rotation to cause a slip-like perturbation by inducing an anterior displacement of the subjects’ Base of Support (BOS) regarding their Centre of Mass (COM) [40]. The belt may reach its maximum reverse velocity at values close to zero [39] or cross the zero speed limit, causing the belt moving anteriorly [40]. Alternatively, Mueller et al. [42] conducted both of these belt-speed profile scenarios. After reaching peak belt speed, the belt’s velocity would return to steady-state walking velocity [40] or zero velocity [39] depending on each study’s experimental protocol. This process mimics overground walking slips that cause a backward LOB. In total, 17 studies provoked slip perturbations to elicit a backward LOB [9,10,11,21,29,30,31,32,33,34,35,36,38,39,40,41,42]. From these studies, Wang et al. [40], Martelli et al. [39] and Lee et al. [41] provoked slip perturbations by sudden acceleration in the forward direction with the treadmill belt operating under unpredictable timing to induce instability to the subjects. Hirata and colleagues [38] applied the perturbations under specific stepping conditions when a subject stepped onto a walkway-mounted treadmill. If the conditions were met, the belt would accelerate from zero speed to maximum velocity and then decelerate at the same rate to zero speed in order to cause a backward LOB, inducing a slip-like perturbation. In this study, the belt velocity did not reverse its direction to cause the perturbations as identified in the trend above, since it was initially stopped before slip onset. On the other hand, study [37] was the only one that considered the application of slip-induced forward LOBs. In this regard, forward-falling slips were provoked by suddenly increasing the treadmill’s belt velocity in the same direction as that used in the steady walking condition. Therefore, there was no reversal of treadmill’s belt velocity as opposed to the studies [39,40,41].

##### Gait Phase Perturbed

Of the 18 studies, 7 provoked slip-like perturbations at the heel strike of the leading foot [9,21,29,37,38,39,42]. The other authors applied the slip during the beginning of the single stance phase [34], initial double-limb support [41], both single- and double-stance phases [40] or shortly after foot touchdown [10,35]. These gait events shortly preceded or immediately followed the heel strike event. Moreover, six studies did not mention the gait event in which the slips were provoked [11,30,31,32,33,36].

##### Gait Speed and Perturbed Leg

Slip perturbations were mostly provoked only on the right or on both legs, in 9 [9,10,11,21,30,31,33,34,37] and 4 [29,38,39,42] studies, respectively.

Furthermore, 50% of these studies instructed their participants to walk at their self-selected speed during the experiments [9,11,30,31,32,35,36,40,41]. While studies [32,35,36,41] instructed participants to ambulate at their comfortable self-selected speed, manuscripts [9,11,30,31,40] required their participants to select a speed from four available options (1.2, 1.0, 0.8 or 0.6 m/s). A third of the papers applied a fixed belt speed across all participants [10,21,33,34,37,42]. Belt speeds of 1.2 and 1.0 m/s were applied in four [10,33,34,37] and two [21,42] studies, respectively. From the three remaining manuscripts, two applied a belt speed that was normalised for each subject according to their leg length [29,39], whereas the other study [38] matched the speed of the subjects to the beat of a metronome, considering both slow (0.9 m/s) and fast (1.6 m/s) walking conditions.

##### Sensor Systems and Data Collection

Most of the studies acquired data from at least an Optical Motion Capture (MoCap) system [9,29,30,31,32,35,37,38]. These data were used to compute spatial-temporal gait parameters [29,38], elevation angles of lower-limb segments [29], upper-limb segment angles [39,41] or stability measures obtained through the computation of COM position and velocity [9,33,34,37,39]. Some authors also collected force data to obtain ground reaction force information [11,21,33,37,38]. Electromyography (EMG) data were collected in two studies [41,42]. Furthermore, Patel et al. [32] and Bhatt et al. [36] collected data from a functional Magnetic Ressonance Imaging (fMRI) system. In these studies, subjects were asked to perform mental imagery of slip events after the perturbation trials.

#### 3.1.2. Overground Walking

Table 3 highlights the 29 studies that delivered slip-like perturbations in overground walking conditions. From these studies, 11 conducted their experiments with only young subjects [10,33,34,46,51,54,55,59,60,61,62], 13 with only older subjects [9,11,12,22,30,31,43,44,45,47,48,49,50], and 3 studies considered both young and older participants [52,53,56]. Additionally, one study allowed the enrolment of young and middle-aged participants (aged between 40 and 60) [57] while another manuscript considered young, middle-aged and older subjects [58]. Only studies [9,11,12,43,44,47,48,49,58] conducted the experimental trials with more than 50 participants. The number of participants ranged from 6 to 195 with a median of 36 subjects.

##### Perturbation Methods

Movable platforms were used in 19 studies [9,10,11,12,22,30,31,33,34,43,44,45,46,47,48,49,50,51,52], whereas slippery contaminant surfaces were the source of instability in 8 other articles [53,54,55,56,57,58,59,60]. Two additional papers described a novel robotic system that applied the perturbations [61,62]. While 19 studies induced slip-induced backward LOBs [9,10,11,12,22,30,31,33,34,43,44,45,47,48,49,50,51,52,62], 9 papers induced LOBs in an unconstrained direction [53,54,55,56,57,58,59,60,61]. Study [46] provoked LOBs in the forward, right and left directions.

Briefly, movable platforms are platforms attached to an aluminum track via ball bearings and are embedded in a walkway [30,45,47]. In regular walking trials, the platforms are firmly locked. However, when a slip trial is about to be conducted, a trigger mechanism releases the device enabling the participant’s leading foot to be exposed to a low-friction surface thus causing a slip. The trigger mechanism focused on the detection of a heel strike event through force plates embedded beneath the movable platforms [30,47]. In a similar approach, Inkol et al. [46] performed a forward translation of the users’ entire support surface by means of a robotic movable platform to provoke slip perturbations. Generally, according to the selected studies, movable platforms elicited slip-like perturbations in the anterior–posterior (AP) direction [11,12,43,48]. Liu et al. [11,12,47] and Sawers et al. [22] mentioned that their platforms could not move in the medial–lateral direction. The platforms were generally installed in pairs, i.e, one on the left and one on the right side of the walkway. The platform of the unperturbed foot was automatically released after the activation of the leading/perturbation platform [43]. In [43,44], after the perturbed platform’s release, the platform of the unperturbed side was released once the recovery foot landed on it.

Another method used to provoke slip perturbations consisted of the application of a slippery surface on a specific location of the walkway. The slippery contaminant solutions were applied to achieve a reduction in COF, which resembles realistic daily-life slippery conditions. Different contaminant solution compositions were used. Merril et al. [53], Nazifi et al. [54] and O’Connel et al. [57] chose a glycerol and water mixture contaminant while Ziaei et al. [55] and Kim et al. [59] selected soapy water as the slippery solution. A single study [56] considered a mixture of water and jelly to induce the imbalance event. Vegetable oil was used to create the slippery surface in [58,60]. With exception for Allin et al. [58] and Arena et al. [60], the contaminant solution was generally applied on top of a force plate to record ground reaction force data in both slip and non-slip trials [54,56,57].

Furthermore, 2 studies applied slip-like perturbations through robotic devices that were connected to the subject during walking and would unexpectedly provoke slips. Rasmussen et al. [61] developed a Wearable Apparatus for Slipping Perturbations (WASP), which consisted of a detachable outsole and a release mechanism controlled via wireless communication. The outsole, worn on subjects dominant foot, initially presented an adequate friction with the floor but, when wirelessly triggered, could elicit a slip-like perturbation during walking by creating a low-friction surface under the foot. This perturbation mechanism was activated by a wireless command sent by an operator in heel strike, mid-stance, or toe-off phases, which had to be anticipated by the operator due to the delay observed between the trigger signal reception and low friction surface release. The WASP was designed to deliver slip-like perturbations unpredictably and unconstrainably concerning the direction and magnitude of the slip. During the trials, beyond the device on the dominant foot which provoked slip perturbations, another WASP outsole (which did not apply perturbations) was used in the non-dominant foot in order to prevent differences in the length of both limbs and ensure a more natural locomotion. In another study, Er et al. [62] developed the Fall Inducing Platform (FIMP), capable of randomly and unexpectedly elicit slip-like perturbations by accelerating subject’s left ankle, while providing freedom of movement. The ankle was interfaced with the perturbation mechanism by a wire rope. The FIMP is programmed with a subject follower algorithm that allows it to automatically follow the subject based on camera footage. It also comprises a gait detection phase algorithm that enables the application of the perturbation on the desired gait event based on data collected by inertial measurement unit (IMU) sensors placed on the subject’s body. There is also an user interface system control that allows the operator to inform the FIMP to trigger a perturbation in the subsequent characteristic gait phase detected, regarding the perturbation chosen to be elicited. In that matter, slips were triggered by forward accelerating the left ankle during the early stance phase by a pull force powered by a DC motor positioned anteriorly to the subject.

##### Gait Phase Perturbed

Sixteen studies [9,22,30,31,33,34,45,46,49,53,54,56,57,58,59,60] provoked the slips at the heel strike. In order to ensure that the provoked slips occurred at this gait event and on the intended slipping leg, slippery solution studies included regular walking trials to guarantee that the slipping foot landed on the zone in which the surface would be contaminated [56]. For instance, O’Connel et al. [57] varied the start position of the walkway to ensure the correct foot placement of the test subjects. Furthermore, study [61] provoked slips at the heel strike, mid-stance or toe-off phases and study [62] elicited slip-like perturbations during the early stance phase. The other 11 studies [10,11,12,43,44,47,48,50,51,52,55] did not mention a specific phase and stated that the slip was elicited when the foot step or contact was detected by the force plates. Nevertheless, this foot contact might be resembled by the heel strike event of the slipping foot in most slip scenarios in the studies’ experiments.

##### Gait Speed and Perturbed Leg

Most studies (16 in 29) applied slips in the right leg [9,10,11,12,22,30,31,33,34,43,44,45,48,49,55,60]. Four studies only perturbed the left leg [53,54,57,62], 5 only the subject’s dominant leg [46,56,58,59,61], and 3 both legs [50,51,52]. One study did not mention the perturbed leg [47]. Moreover, 18 manuscripts instructed their participants to ambulate at their self-selected speed [9,10,11,12,33,45,46,47,48,49,53,54,55,56,57,58,61,62], 3 studies matched the speed of the subjects to the beat of a metronome [50,51,52] and study [60] instructed participants to maintain their speed between 1.45 and 1.60 m/s. The remaining 7 studies did not mention the speed instructed during the trials [22,30,31,34,43,44,59].

##### Sensor Systems and Data Collection

All the studies used an Optical MoCap for data acquisition [9,10,11,12,22,30,31,33,34,43,44,45,46,47,48,49,50,51,52,53,54,55,56,57,58,59,60,61,62]. Data from these systems were used to compute spatial-temporal gait parameters [56,61], lower limb segment angles [49], joint angles [59,62] and joint moments [44,49] or stability measures obtained through the computation of COM position and velocity [43,44,46,47,51,52]. Additionally, 22 studies also included force plates to collect ground reaction force data [9,11,12,22,30,31,33,34,43,44,45,46,47,48,49,53,54,55,56,57,58,59]. EMG sensors were included in 5 manuscripts [22,45,54,56,57] and 3 studies used wearable inertial sensor systems [56,60,62].

### 3.2. Trip-like Perturbations

Trip-like perturbations were conducted in 15 studies. From this group, 6 papers described the application of a trip during treadmill walking, whereas the remaining 9 articles concerned its delivery in overground walking conditions. Figure 3 presents some of the methods used to provoke trips in the selected studies.

#### 3.2.1. Treadmill Walking

Table 4 shows the methods applied to induce trip-like perturbations during treadmill walking. Four studies involved young adults [41,42,63,64] and one study included middle-aged adults [65]. Study [66] enrolled young, middle-aged, and older adults. All the studies conducted the experiments with less than 25 participants. The sample size ranged from 8 to 24 participants with a mean number of 14 subjects, approximately.

##### Perturbation Methods

Table 4 shows that half of the studies (3 studies) induced trip-like perturbations during treadmill walking by suddenly changing the belt’s acceleration. In addition, 2 studies elicited trips using a brake-and-release system connected to the subject’s leg. The remaining study used an external device to unexpectedly place a tripping board to perturb subjects’ locomotion. All of these studies provoked trip-induced LOBs in the forward direction [41,42,63,64,65,66].

Concerning the studies that provoked trip perturbations by changing the treadmill’s belt acceleration, Lee et al. [41,63] elicited the trips to the non-dominant leg by the sudden stoppage of the perturbed foot treadmill belt. After the detection of the first heel strike from the unperturbed foot following the perturbation, the belt returned to its pre-perturbation speed, which allowed the subject to recover from the trip-like event by continuing walking. Mueller et al. [42] provoked trips to both legs by accelerating the treadmill belt shortly after the heel strike event. This accelerating period was followed by a deceleration phase towards the pre-perturbation speed of 1 m/s.

In a different approach, brake-and-release systems have also been used in 2 studies. These systems unexpectedly apply resistance to the subject’s gait and inhibit the foot from going forward, emulating a trip event. In regular walking without perturbations, these systems accompany subject’s walking without hindering it [64,65]. König et al. [65] applied resistance to the right foot during the swing phase via an ankle strap which was pulled through the brake-and-release system with a Teflon cable [71]. This device was able to generate a force around 55 N with a rise and fall times of the pulling force under 20 ms. During non-perturbation trials, the resistance received by the participants was below 0.1 N, which was considered negligible [65,71]. The cam-based mechanism developed by Aprigliano et al. [64] was also able to stop the forward motion of the right foot along with the swing phase. During unperturbed walking, the rope moved according to the foot’s movement. A nylon rope was attached to the participants’ foot at one side and to the main frame of the brake-and-release system on the other side by a compliant spring with a stiffness of 3 N/m.

The remaining study proposed a tripping device. Silver et al. [66] included a device capable of unexpectedly placing an obstacle in front of a subject during single-belt treadmill walking. This tripping system randomly delivered one of 2 selected objects to the left foot, one closed and one opened, both matching in volume and with a parallelepiped shape. The closed object was used to elicit a perturbation similar to a trip over a solid object placed on the floor while the opened one was used to mimic a trip event where the foot of a subject is “caught” by the open area underneath the obstacle.

##### Gait Phase Perturbed

Three studies delivered the trip-like perturbations during the swing phase [64,65,66], two at the initial double-limb support phase [41,63], and one at the heel strike [42].

##### Gait Speed and Perturbed Leg

While 3 studies instructed their participants to ambulate at their self-selected speed [41,63,66], the other 3 applied a fixed speed across all the subjects [42,64,65]. Two studies applied the trip-like perturbations to the right leg [64,65], 2 to the subjects’ non-dominant leg [41,63], 1 to the left leg [66], and the remaining one to both legs [42].

##### Sensor Systems and Data Collection

Optical MoCap systems were used to collected data in 5 studies [41,63,64,65,66]. Data from these systems were used to compute COM position and velocity, as well as upper body segment angles [41,63]. Force data were acquired through force plates embedded on the treadmill in [41,63] or by using a plantar pressure insole in [42]. EMG sensors were used in [41,42]. Wearable inertial sensors were considered in both brake-and-release systems studies [64,65].

#### 3.2.2. Overground Walking

Table 5 presents the 9 studies that conducted trip-like perturbations during overground locomotion. Five studies conducted experiments with only young adults [51,60,62,67,69], 2 manuscripts with only older adults [50,68] and 2 studies with both young and older adults [52,70]. All the studies conducted the experimental trials with less than 50 participants. The number of subjects ranged from 6 to 44 with a median of 12 participants.

##### Perturbation Methods

Regarding the group of 9 studies that performed trip-like perturbations during overground walking, the triggering of a tripping object was the main source of instability. More specifically, this tripping object was activated in 2 different ways, either by the automatic spring of the obstacle from the floor (6 studies [50,51,52,60,67,68]) or by the manual placement of the object in the walkway (2 studies [69,70]). In a different approach, Er et al. [62] considered the application of trips by a novel robotic device.

Concerning the studies that described the obstacle triggering, Potocanac et al. [67] included a walkway with a layout of 14 hidden obstacles (15 cm height) arranged in a row. During perturbation trials, any of these obstacle could be released from the floor to cause a trip. The released obstacle was selected by a kinematic data-based algorithm while the subject was approaching the obstacles zone [67,72]. In Okubo et al. [50], 14 cm height trip boards could suddenly flip up from the walkway to cause a trip. These obstacles were triggered by a foot detection sensor during perturbation trials and were released 50 ms before the foot arrived at the hidden board position. This would lead to the automatic delivery of perturbations in the mid-swing phase. If participants reported increased levels of anxiety or perceived difficulty during the trials, the trip board height was decreased to 7 cm. Wang et al. [68] applied an unexpected trip event by releasing an 8 cm height hinged metal plate obstacle in less than 150 ms. This board was locked by the powered electromagnets, but when the unperturbed limb’s ground reaction force measured by a force plate (placed before the hidden obstacle) exceed 80% of the subjects’ body weight, the electromagnets were turned off and the plate was unlocked to elicit a trip. During baseline non-perturbed walking trials, the starting position of the test subjects was adapted in order to assure that trip-like perturbations were applied during the mid-to-late swing phase [68]. Furthermore, in studies [51,52], 14 cm height tripping boards sprang up from the walkway in order to elicit trip events on the subjects. During the first trials, Okubo et al. [52] used a 7 cm height board and further increased the height to 14 cm once the participant became more confident that they could avoid falling. However, in both of these studies, the obstacle was wirelessly released by a trained tester when the participants leading foot passed beside the location of the hidden trip board, such that the perturbation occurred at the mid-swing phase. Unlike the first 3 mentioned studies [50,67,68], in [51,52] the trip obstacle was manually triggered. Although Arena et al. [60] mentioned that the trip obstacle embedded in the walkway was manually actuated, it is not clear if that activation is performed remotely as in [51,52].

Two studies manually placed the obstacles in the walkway. Ko et al. [69] and Schulz [70] placed objects in the walkway to induce trip-like perturbations during the unperturbed limb stance and perturbed limb swing phases, respectively. In [70], the obstacles were either visible or hidden. In the visible layout, the obstacles were white coloured placed on a black surface, whereas in the hidden layout both the obstacles and the surface were black.

The remaining study considered the use of FIMP, a robotic device that elicits trip-like perturbations [62]. The device allowed freedom of movement in non-perturbation trials and was able to posteriorly arrest the participant’s left ankle with an electromagnetic brake to cause an unexpected trip. Subject’s ankle was connected to the FIMP’s brake system by wire ropes.

##### Gait Phase Perturbed

All the studies considered the application of the trip-like perturbations during the swing phase to provoke trip-induced LOBs at the forward direction [50,51,52,60,62,67,68,69,70].

##### Gait Speed and Perturbed Leg

Studies instructed the participants to match their speed to the beat of a metronome [50,51,52], adopt a variable walking speed [60] or to walk at a self-selected speed [62,67]. Schulz [70] instructed participants to walk under three different speed conditions, namely slower than preferred speed, preferred speed and as fast as safely possible speed. Ko et al. [69] did not mention the speed instructed. Moreover, three studies perturbed both legs [50,51,52], three perturbed the right leg [60,67,69], and two perturbed the left leg [62,68]. Study [70] did not mention which leg was perturbed.

##### Sensor Systems and Data Collection

Optical MoCap systems were used for data collection in all of the mentioned studies [50,51,52,60,62,67,68,69,70]. These systems provided data to compute the COM position and velocity [50,68], spatial-temporal gait parameters [67,68], and upper-body segment angles [68]. Force data were acquired in [67,68]. An EMG system was considered in [67] and wearable inertial sensors were used in [60,62].

### 3.3. Methods Used to Unbias the Perturbations

From the group of 48 selected studies, some strategies were adopted to mitigate participants’ anticipatory behaviours towards the perturbations applied.

Some studies attempted to affect participants’ vision in order to reduce the likelihood of them predicting the position of the obstacles and the perturbations’ occurrence. Studies instructed subjects to look straight ahead while walking [32,35,53,57] or to fix their sight on an object positioned at eye level [41,59]. Other authors dimmed the lights to prevent visual cues that would allow participants to identify potential slippery areas [53,54,57,58,60] or tripping obstacles [70]. Silver et al. [66] and Ko et al. [69] attempted to occlude subjects’ peripheral visual field with special goggles and eye patches, respectively. Studies also instructed participants to face away from the walkway before each trial to limit their perception regarding positioning of the slip or trip perturbation sources [51,53,57,58]. As such, the perturbation sources could be added, removed, or moved to different places along the walkway between trials [50,51,52,69].

Furthermore, studies did not inform participants about the perturbations’ characteristics and provoked perturbations with different: (i) intensities [29,32,39]; (ii) directions [46]; (iii) gait events perturbed [61,62]; and (iv) trial time length [61]. In addition, Okubo et al. [50,51] applied slips and trips in a mixed order. Moreover, other studies provoked perturbations to both legs [29,38,39,42,50,51,52].

Some authors only applied a single perturbation in order to minimise gait adaptations following repeated perturbation exposure [37,59,60]. Studies also conducted walking trials without perturbations between perturbation trials in order to increase the unpredictability of perturbation delivery [50,68].

## 4. Discussion

There was a greater prevalence of slip-like perturbation studies than trip-like perturbation studies. This is in line with the evidence detailing that slips contribute more to fall events than trips [24,25].

From the group of 48 selected studies, 22 enrolled young subjects [10,21,32,33,34,36,37,38,41,42,46,51,54,55,59,60,61,62,63,64,67,69], 16 included older participants [9,11,12,22,30,31,35,40,43,44,45,47,48,49,50,68] and 6 accounted for both young and older subjects [29,39,52,53,56,70]. The remaining studies enrolled young, middle-aged and older adults [58,66], young and middle-aged participants [57], and middle-aged adults [65]. The high number of studies that included participants of young and middle-aged groups motivated this review to extend the analysis of the review study [3]. Moreover, only 9 out of the 48 selected studies included more than 50 participants during the experimental trials [9,11,12,43,44,47,48,49,58]. These search results depict the prevalence of young subjects during the experimental protocols for provoking artificial falls, as well as the low number of participants enrolled. Considering that the elderly bear a substantially higher fall risk than younger adults, efforts should be made to include older participants in future experiments. Additionally, a greater number of participants should be enrolled in order to allow generalisation of the study’s findings over a greater sample of the population. These efforts would promote a better and wider understanding of the reactions of high fall-risk subjects to slip and trip gait perturbations.

When applicable, the search results obtained from this narrative review will be compared against the evidence reported in the review studies conducted by McCrum et al. [3] and Karamanidis et al. [23].

### 4.1. Which Methods and Walking Conditions Are Used to Provoke Slip- and Trip-like Perturbations?

This review finds a variety of procedures implemented in the scientific literature to elicit slip- and trip-like perturbations.

Slip-like perturbations were applied during both treadmill and overground walking with the latter (29 studies) being more prevalent than the former (18 studies). The application of slips during treadmill walking consisted of a sudden change in the belt’s acceleration to induce anterior displacement between the BOS and the COM. This finding is aligned to that by Karamanidis et al. [23]. On the other hand, overground walking slips were delivered by movable platforms, slippery solutions, and novel robotic devices. McCrum et al. [3] had also found the prevalent use of movable platforms to provoke slip perturbations.

The slip-like events were generally applied at the instant of the heel strike. This is in line with Lockhart [20] who highlighted this gait event as the most hazardous one towards slip events during walking. During the heel strike, the individual’s body weight is transferred to the limb where the heel strike is taking place (slipping limb). Applying a slip to this limb when the transfer is incomplete causes a highly unstable situation due to inadequate and stable body support provided by the lower limbs. Additionally, Lockhart [20] also highlighted the toe-off event as another critical gait phase for slipping. However, since in this event almost all of the body weight has been transferred from the toe-off limb (slipping limb) to the other limb (trailing limb), the likelihood of inducing a hazardous situation is smaller than the slip perturbation at heel strike [20]. This may be the reason why only one study applied slips at the toe-off phase [61].

Although overground walking conditions are more realistic to emulate daily life locomotion, treadmill devices provide continuous collection of gait patterns over longer periods of time. Thereby, slip perturbation’s onset timing during treadmill locomotion may be more unpredictable than during overground walking [73]. Regarding real-world slips, treadmill slips may not resemble the whole nature of slips, since treadmill belts can often move in only one direction, generally considered the AP direction [21,29,38,39]. Despite this, study [74] shows that from the total of the instability-induced falls collected during their experimental protocol, only 8.2% were related to medial–lateral (ML) direction. Hence, there is evidence from the literature that supports the AP direction as the most relevant in slip dynamics, which may provide support to studies that performed slip-like perturbations during treadmill walking.

Moreover, concerning real-world slips, the slippery solution-based perturbations are more likely to resemble real slippery conditions by reducing COF at the interface between the foot and the floor [20]. Herein, the perturbation direction is not restricted which allows for more realistic and unpredictable slip dynamics as they happen in real life. However, the slip direction and magnitude unpredictability may hinder the generalisation of specific findings of slip dynamics, since the slip conditions are less controlled. When the perturbation is controlled, i.e, its characteristics can be normalised across different experimental trials, specific aspects of the slip dynamics and responses can be studied which can enable a better and more reliable generalisation of the studies’ findings [39]. In addition, the slippery solution studies did not automatically deliver the perturbations, which could yield more variability on the onset and the magnitude of the provoked slip. Conversely, slips caused by movable platforms, novel robotic devices and by changing the treadmill’s belt acceleration were delivered automatically.

Trip-like perturbations were also elicited during treadmill or overground locomotion. Treadmill walking trips were caused by changing the belt’s acceleration, using a brake-and-release system or a tripping device. Karamanidis et al. [23] verified the same methods except for the tripping device. On the other hand, overground walking trips were caused by triggering an obstacle release, manual placing an object along the walkway, or using a novel robotic device. Trips were mainly applied during the swing phase of gait. This is in line with Karamanidis et al. [23], where they described trip events as a sudden disruption in the relation between subject’s COM and BOS caused by the abrupt interruption of the swing limb motion. The advantages and disadvantages of using treadmill or walkway setups to provoke trips are analogous to those mentioned above for the slip-like perturbations. All the treadmill-based setups conceived to apply trip-like perturbations, except study [66], did not use objects to perturb the locomotion. Thereby, the perturbations were applied either by changing the belt’s acceleration or brake-and-release systems may be less likely to resemble real-world trips. Nevertheless, the latter systems are able to interrupt the swing phase of gait by directly pulling the respective foot, which can also accurately depict trip events. Meanwhile, in the manual obstacle placement it was not guaranteed that the trips would occur at a specific phase of gait, and studies that triggered an obstacle release had a more automatic and reliable way to perturb participant’s gait during their swing phase [50,51,52,67,68]. Moreover, regarding the tripping obstacles implemented in the selected articles, the prominent use of boards with height values of 7/8 cm [60,68] and 14/15 cm [50,51,52,67] was observed. Predictably, an inter-trial variability in perturbation onset may arise in the obstacle manual trigger. Therefore, it is recommended that researchers consider more automatic approaches to deliver trips. It is worth noting that from all the trip-like perturbation studies, only Silver et al. [66] considered the application of more than one type of obstacle. Future work should follow this approach in order to enable a more comprehensive analysis of the variability of trip events triggered by different types of obstacles.

For both perturbation types, the selection of walking condition (treadmill or overground) must account for the trade-off between the relevance of inducing natural perturbation dynamics and the generalisation of the studies’ results. Further, the selection should attend to the available space for the experimental setup and whether the perturbations are intended to be delivered: i) automatically or not; and ii) directly from a device connected to the subject or not [61,62,64,65,65]. Additionally, for slip perturbations, this selection has to also bear in mind whether the perturbations are intended to be delivered in a specific direction or not. For trip perturbations, it is also necessary to consider whether one or more types of obstacles are planned to be used to apply the trips.

### 4.2. Is It Preferable to Deliver Perturbations during Treadmill or Overground Walking in PBT?

The results showed that there is a prevalence of overground walking perturbations relative to treadmill perturbations in both slip- and trip-like perturbations. The walkway perturbations may be more realistic than the treadmill-based ones, since real-life slips and trips occur during overground walking. However, recent research has been attempting to validate the perturbation delivery during treadmill locomotion against overground walking [9,11]. Liu et al. [11] compared the retention of fall-resisting skills in the follow-up period of 6-months between treadmill and overground walking conditions. Results showed that the group of individuals that received baseline overground walking perturbation training had a lower fall incidence and a higher reactive stability against an overground slip applied 6-months after the perturbation training sessions than the group that underwent the baseline treadmill perturbation training. Therefore, since the treadmill slip training group achieved a lower balance recovery performance than the overground slip training group, the authors could not generalise the delivery of slips during treadmill walking against overground walking. Nevertheless, the treadmill perturbation training group had increased stability metrics and lower fall incidence than its control counterpart (treadmill training without perturbations), demonstrating the long-term relative retention of fall-resisting skills from treadmill perturbation training.

Researchers are working on this topic given all the advantages associated with treadmills to provoke perturbations. Perturbation delivery during treadmill walking provides accurate velocity profile control, which may lead to the reliable delivery of slips [30]. Computer controlled treadmill devices enable the application of different perturbations in a highly precise and standardised manner, which is not observed during overground perturbations studies since the perturbation characteristics entail more variability [40,42]. Additionally, treadmill devices allow researchers to easily modulate the intensity of the perturbation and, therefore, to study subject’s adaptation to different perturbation characteristics [40]. Furthermore, the portability of treadmill devices, as well as the reduced space they occupy are also noteworthy [30,34,40]. Treadmills also allow for the collection of multiple and continuous walking patterns over long periods of time [73]. As reported in previous review studies [3,23,26], this will also lead to an increased difficulty to predict when the perturbation will be delivered, which ensures more realistic reactive balance control strategies adopted by the participants. Concerning all of the above mentioned advantages of treadmill perturbation delivery, the generalisation of treadmill perturbation applications against those applied during overground walking becomes essential [34,40].

### 4.3. Is It Preferable to Use a Single-Belt or a Split-Belt Treadmill to Perturb Walking?

Both single- and split-belt treadmills have been used to apply treadmill gait perturbations. Single-belt treadmills were used in 14 studies [9,10,11,30,31,32,33,34,35,36,40,64,65,66], while split-belt treadmills were adopted in 8 manuscripts [21,29,37,38,39,41,42,63]. Compared to single-belt, split-belt treadmills give researchers the opportunity to study kinetic data from both feet independently by integrating force sensors in each of the belts. Additionally, the application of gait perturbations is more standardised and reproducible across different test subjects, allowing to define: i) more accurately the limb that is going to be perturbed; ii) specific velocity profiles for each belt to conduct the perturbation; and iii) automatic onset of the perturbation based on kinetic data from the targeted limb [39,42]. These features render the split-belt treadmill more suitable for delivering realistic perturbations. However, a comparison study [73] of the walking kinematics between the single- and split-belt treadmill walking, concluded that subjects tend to widen their base of support while walking on a split-belt treadmill to manage the walking gap between the two belts. Despite this unnatural gait pattern, frontal plane lower-limb kinematics were not significantly different between both treadmill types [73]. With this in mind, it is possible to deliver perturbations on a split-belt treadmill taking advantage of its features compared with its single-belt counterpart. Regardless, single-belt treadmills are more accessible concerning the application of gait perturbations [32,35,64,66].

### 4.4. What Procedures Are Implemented to Maintain Responses to Perturbations Unbiased?

Different mechanisms were adopted by the authors to reduce the predictability of the perturbation events delivered. Generally, all the studies detailed that perturbations were intended to be unexpected and instructed their participants to react naturally and try to recover balance whenever a perturbation was applied. As such, studies did not conduct trials to familiarise subjects with the perturbations to mitigate participants’ learning effects and gait adaptation to the perturbations. Previous literature studies suggested that following the first perturbation exposure, subjects alter their gait characteristics to adapt to those perturbation conditions [47,75]. Thus, some authors tripped or slipped their participants only once [12,36,37,47,53,54,57,58,59,60,65].

However, 38 studies [9,10,11,21,22,29,30,31,32,33,34,35,38,39,40,41,42,43,44,45,46,48,49,50,51,52,55,56,61,62,63,64,65,66,67,68,69,70] conducted multiple perturbations on the same subject. Consequently, they included different techniques to enhance the unpredictability of both the perturbation’s onset and characteristics so as to limit the bias associated with the subjects’ gait modification following repeated perturbation exposure and increase the reliability of the results obtained. Some authors conducted perturbations at different intensities [29,32,39], directions [46], and on both limbs [29,38,39] to prevent subjects’ adaptation to only one type of perturbation. Other sources of variability included changing the gait phase to be perturbed [61,62], trial duration [61], and the location of obstacles [51]. Overground walking perturbation studies often report that, between different trials, participants were required to face away from the walkway in order to limit their perception of the perturbations source’s placement [51,58,69]. Furthermore, most of these studies included non-perturbation trials between perturbation trials to create more unexpected conditions when perturbations were delivered.

Some authors attempted to somehow manipulate the participant’s field of view to decrease the likelihood of them perceiving the perturbation onset. Studies instructed their subjects to look straight ahead while walking [35,53,57] and focus their sight on an object [59] or a mark [41] at eye level. Silver et al. [66] and Ko et al. [69] presented another approach to limit participants’ peripheral vision by requiring them to wear an eye patch during the trials. In slip-like perturbations through slippery solutions, authors created a dimmed-lighting environment so as to prevent participants exploiting light reflections arising from the liquid surface to spot its location [53,54,58]. Furthermore, Schulz [70] created a low-lighting environment that would reduce subjects’ ability to spot tripping obstacles. However, since the introduced visual constraints do not realistically reflect real-life walking, its application should be avoided to mimic daily-life conditions.

It is also worth noting that any gait asymmetries provoked by the placement of some constraint on the subject’s body during the trials must be compensated to ensure a natural gait. In this regard, despite the fact perturbations were applied to only one limb, participants in [61] wore an outsole device on both feet and in [64] a rope was attached to each foot.

### 4.5. Which Limb Is Generally Used to Apply the Perturbations?

Studies have applied the perturbations on one or both legs. However, some works did not mention which leg or legs were perturbed [32,35,36,40,47,70]. Most of the studies (35 studies) applied the perturbation to only one leg. From this group of articles, 22 and 8 papers applied the perturbations to the right and left legs, respectively, whereas the 5 remaining studies described perturbation delivery to the dominant leg. It is also noteworthy that from the eight studies that only perturbed the left leg, two of them described that leg as the participant’s non-dominant leg. There is clear evidence for the preferential application of perturbations to the right limb instead of the left. This may be related to the fact that the large majority of individuals present right-side dominance [76]. However, only seven studies have reported on the limb selection according to the side-dominance of individuals: the dominant [46,56,58,59,61] or the non-dominant limb [41,63]. Additionally, Okubo et al. [50,51,52] also considered the side-dominance of the subjects, despite the fact the authors conducted perturbations on both legs. Despite side-dominance being overlooked by most of the collected studies (works [44,57] reported this limitation), it should be considered, especially if the study considers PBT, where perturbations are delivered to improve subjects’ balance recovery skills. Martelli et al. [77] highlighted that the dominant limb is mainly responsible for propelling the body forward, whereas the main role of the non-dominant limb is to provide body support. A previous work also suggests that there is an increased risk of falling in situations where the perturbed limb is the non-dominant limb [78]. Therefore, the analysis of subjects’ reactions to perturbations is more comprehensive when leg dominance is also considered.

Furthermore, seven studies perturbed both legs of participants [29,38,39,42,50,51,52]. Although this represents less than 15% of the included studies, perturbation delivery to both legs plays an important role in maintaining natural responses to perturbations from the participants. As mentioned by Martelli et al. [39], Aprigliano et al. [29] and Mueller et al. [42], although only data from right-sided perturbations were considered for the study’s analysis, left-sided perturbations were also randomly delivered to obtain unbiased results by limiting subject’s gait pattern adaptation to only right-sided perturbations. The randomisation of the perturbation side allows researchers to decrease learning effects and thus to counteract the induced LOBs and participants’ awareness of the perturbations’ characteristics in comparison with perturbations provoked only to one side [38,50,51,52]. Moreover, the application of perturbations to both sides would enable the differences between subject’s reactions to perturbations applied to the dominant and non-dominant limbs to be studied. It is also noteworthy that more complex resources are required in order to deliver perturbations to a specific limb. A split-belt treadmill is more suitable to reliably apply perturbations to only one limb during treadmill walking [33]. However, these treadmill devices are less available in the market regarding the single-belt treadmill. Similarly, concerning overground walking perturbations, the walkway should be divided into independent segments for each limb so as to enable reliable and reproducible perturbation delivery towards a specific leg [79].

### 4.6. Which Was the Participants’ Walking Speed during the Trials?

From the 41 studies that included trials under only one walking condition, either treadmill or overground locomotion, subjects were instructed to walk at self-selected speed in 23 studies [12,18,32,35,36,40,41,45,46,47,48,49,53,54,55,56,57,61,62,63,66,67], 10 studies [21,37,38,42,50,51,52,60,64,65] applied a fixed walking speed for all participants, 2 studies [29,39] described the application of a normalised walking speed specific to each subject, and 6 studies [22,43,44,59,68,69] did not mention the walking speed adopted. The seven studies that delivered perturbations during both treadmill and overground walking, reported different velocity paradigms for each walking condition. For instance, Yang et al. [10] and Lee et al. [33] described the application of 1.2 m/s treadmill speed and a self-selected speed while walking along the walkway. The other studies either reported self-selected speed in both walking conditions [9,11] or did not describe the overground walking speed [30,31,34].

Regardless of the walking condition adopted, most of the authors instructed their participants to walk at a self-selected speed. This is in line with the review conducted by McCrum et al. [3]. Regarding these 23 manuscripts, subjects selected their own comfortable speed in the overground walking studies [12,18,45,46,47,48,49,53,54,55,56,57,61,62,67], while in treadmill walking studies participants were instructed to either ambulate at their own self-selected speed [32,35,36,41,63,66] or were asked to select a speed from four available options (1.2, 1.0, 0.8 or 0.6 m/s) [40]. Although walking at a comfortable speed would simulate more realistic walking conditions, it is more difficult to deliver perturbations that are equally challenging across all the subjects.

Studies [21,37,42,64,65] selected a constant speed throughout all the trials towards mitigating the problem associated with the different velocities from the different participants. In these manuscripts, belt speeds of 1.0 [21,42,64], 1.2 [37], and 1.4 [65] m/s were applied. The constant speed condition is simpler to perform in the treadmill than in overground since a constant belt speed can be employed. However, some overground walking perturbation studies adopted strategies to overcome the limitation. Arena et al. [60] initially instructed their participants to walk naturally while monitoring their gait speed using Optical MoCap data. Afterwards, participants were told to increase or decrease their speed maintaining a velocity between 1.45 and 1.6 m/s. In addition, Okubo et al. [50,51,52] regulated participants’ speed using a metronome such that they stepped on tiles positioned along the walkway in time with the metronome beat. These tiles were configured according to subject’s cadence and normal step length. Hirata et al. [38] also matched the speed of the subjects with the beat of a metronome, considering both slow (0.9 m/s) and fast (1.6 m/s) walking conditions. However, in this constant walking speed condition, subjects’ normal walking conditions are partially disregarded and the velocities in which perturbations are delivered may not still be equally challenging for subjects with different anthropometric characteristics [37]. In order to overcome this limitation, Martelli et al. [39] and Aprigliano et al. [29] calculated a velocity that was specific to each participant according to their leg length. This procedure is in line with McCrum et al. [80], who claimed that the walking speed should be adapted to each subject to induce a similar margin of stability across all participants and trials. This procedure enables researchers to create an equally challenging environment of perturbation delivery regardless of the subjects’ physical capabilities.

### 4.7. What Are the Main Sensor Systems Used to Collect Data during Perturbation-Based Protocols?

The Optical MoCap devices were the most used sensor systems (45 out of 48 studies). The use of reflexive markers to acquire subject’s motion data enables the extraction of relevant kinematic and angular information of subjects’ motion in laboratory conditions. This may be particularly important to find parameters that relate to the biomechanical reactions to falls, which could be employed in the development of fall prevention strategies for individuals with increased risk of falling, as shown in [39,48,49]. Data from these systems were used to compute subject’s stability through the computation the COM position and velocity [9,34,37,39,41,44,46,47,51,68] and for the analysis of spatial-temporal gait parameters [29,38,56,67,68], upper limb segment angles [39,41,68], lower limb segment angles [29,49], joint angles [59], and joint moments [44,49].

Force data were acquired in 32 studies. These data were mainly collected by force plates either installed beneath treadmill’s belts [38,39] or embedded in some position along a walkway [44,56]. The main purpose for using the force plates was to provide ground reaction force information to detect the reliable timing of perturbation application rather than to collect data for further analysis [39,42,43,68].

In five studies, participants were equipped with wearable inertial sensors serving different purposes. Arena et al. [60] placed an IMU on the forehead to acquire meaningful information of head motion during slip and trip events. Aprigliano et al. [64] collected inertial data to compute several limb elevation angles during normal and perturbed walking trials. Additionally, Er et al. [62] used data provided by inertial sensors to feed a gait event detection algorithm that led to a precise application of the gait perturbations.

Lastly, EMG sensors were considered in eight studies. They were the only type of biosensor used among the 48 studies collected. EMG data may provide useful information about muscles activated during imminent fall-risk situations [81]. Accordingly, Sawers et al. [22,45] and Nazifi et al. [54] explored muscle synergies, which represent groups of muscles that coactivate to produce a biomechanical function required to perform a certain motor task [82], during perturbation trials. The study of the synergies underlying gait perturbation recovery could be promising in better understanding which muscles are not being properly activated in fall-risk individuals. In addition, it can also promote an evidence-based fall prevention treatment. Moreover, during human imbalance condition, a sudden EMG pattern alteration due to a reactive neuromuscular response may be generated faster than the modification of inertial signal patterns [83]. Accordingly, Marigold and Patla [84] and Pijnappels et al. [85] showed that rapid lower limb muscle activation was elicited following slip- and trip-like events, respectively. Thereby, future studies should consider the use of EMG sensors for the relevant information they provide towards the better understanding and faster detection of human’s reactions to perturbation events.

Although load cells were used in some studies, these sensors did not provide meaningful data regarding the subject’s reactions to provoked perturbations since their main purpose was to label the experimental trials as fall or non-fall/recovery [30,47,51].

### 4.8. Are There Benefits to Apply Both Slip- and Trip-like Perturbations?

Among the included studies, six delivered both perturbation types to each participant [41,42,50,51,52,62]. The application of both perturbation types may be more suitable than the single type of perturbation if the purpose of the study concerns PBT [41,42,51,52]. Okubo et al. [51] pointed out that the application of only slip-like perturbations may lead individuals to learn recovery strategies related to the predictive adaptation of anteriorly shifting the COM, which may in turn increase the risk of tripping. Thus, to adapt individuals to real-life perturbations these COM predictive alterations must be mitigated by the mixed application of both slips and trips. Furthermore, study [52] applied slips and trips in a mixed order to mimic more realistic perturbation conditions. This would ensure more natural and realistic reactions to the gait perturbations applied. Nevertheless, the inclusion of perturbation types may yield a more complex experimental protocol.

### 4.9. What Is the Current Feasibility of PBT Implementation in Clinical Settings?

The emerging PBT paradigm towards fall prevention has shown encouraging results in fall-rate reduction in recent years. Some experimental studies included in this review [12,40,68] reported that PBT sessions allowed older adults to develop short- and long-term retention of fall-resisting skills, such as improving their feedforward and feedback adjustments of COM stability and body kinematics to unexpected perturbations. These findings are in line with the review conducted by McCrum et al. [26]. Liu et al. [12] observed that older subjects fell significantly less due to slips provoked after receiving PBT at 6-, 9-, and 12-month follow-ups compared to the single slip applied during perturbation training. Furthermore, studies [11,50] verified that participants who enrolled in a PBT session experienced fewer falls and had improved postural stability for perturbation recovery than participants from the control group. Okubo et al. [52] verified that after the perturbation training completion, older adults were able to improve their margin of stability for balance recovery, which reduced their slip-induced falls from 28.6% to 14.3%. Furthermore, Wang et al. [68] reported that despite the fact that 48% of the older adults included in their study fell on the first trip, no participant fell on the trip provoked after perturbation training.

Despite these promising results, there still remains a lack of standardisation regarding PBT protocols conducted in the scientific literature. The number of perturbations within each PBT session ranges from 1 perturbation [12] to 40 perturbations [9,11]. Sessions are usually completed within one [11,52] or two days [50]. In addition, the retention of fall-resisting skills is either analysed immediately after the PBT session [11,40,50,52,68] (after 30 min [68] or 1 hour [50]), or in long-term follow-up periods of 6 [11,12,31], 9 [12] and 12 months [12]. This lack of standard PBT protocols hinders the implementation of PBT in clinical practice [26].

As portrayed by McCrum et al. [26], standard perturbation doses must be established. Accordingly, further research must clarify the optimal number of PBT sessions, the number of perturbations delivered within each session as well as the perturbation intensity profile during each session in order to maximise the long-term retention of the fall-resisting skills acquired during the training. In this regard, McCrum et al. [26] acknowledge the need to perform dose–response studies to uncover these optimal training conditions, which should be preferably conducted using the more clinically appropriate treadmill-based protocols. Currently, the perturbation dosage which yields the maximum efficacy remains unknown [26]. Lee et al. [31] observed that from a treadmill PBT protocol including a lower (24 slips) and a higher (40 slips) perturbation dosage groups, only the higher dosage group showed a significantly better stability performance than the control group in the 6-month follow-up overground slip perturbation. However, Lee et al. [30] findings suggest that a higher practice dosage (40 slips) did not necessarily benefit the performance of older subjects to counteract an overground slip after a treadmill PBT session in comparison with a lower practice dosage (24 slips). While Lee et al. [31] concluded that the higher perturbation dosage might be critical for the long-term retention of fall-resisting skills, findings by Liu et al. [12] suggest that a PBT session containing a single slip may be effective to prevent future falls. As such, future research must address this issue to define a standard perturbation dosage to include in PBT sessions.

Furthermore, PBT requires additional safety measures compared to conventional balance training [26]. Safety harness systems are frequently used to prevent falls during PBT sessions. These systems guarantee the subject’s safety by arresting any irreversible LOB that would lead to a fall while assuring subject’s freedom of movement during regular gait and allowing the therapist to remain focused on perturbation delivery. This equipment is usually ceiling-mounted with a low-friction trolley-and-beam system either above the treadmill [9,11] or the overground walkway [9,11,12,52,68]. The safety harness system is required to be comfortable and well-fitted as to not interfere with participants’ biomechanical responses to the perturbations and prevent discomfort during and after training [26,50].

In addition, it is mandatory to objectively assess the anxiety levels exhibited by participants during PBT sessions, as the increase in perturbation intensity and unpredictability may lead to participant withdrawal. Accordingly, Okubo et al. [52] pointed out the need for further research to investigate and develop multi-day PBT interventions that are suited to individuals’ ability to counteract perturbations and anxiety levels. It is also worth noting that the costs related to the currently used perturbation systems as well as the expert personnel required for their operation may hinder the implementation of PBT in clinical settings [3].

Therefore, new experimental studies should attempt to fill the current gaps in order to provide clinicians with guidance and develop standard clinical practice guidelines for PBT implementation [26].

## 5. Conclusions

This narrative review analysed the current *state-of-the-art* research on artificial slips and trips perturbation delivery.Slip perturbations were provoked during treadmill walking by changing the belt’s acceleration or during overground locomotion by using: (i) a movable platform; (ii) a slippery solution; or (iii) novel robotic devices. Trips were provoked during treadmill locomotion by: (i) changing belt acceleration; (ii) using a brake-and-release system; or (iii) using a tripping device. Overground trips were elicited by: (i) triggering an obstacle; (ii) manually placing an obstacle along the walking path; or (iii) using a novel robotic device. Despite most studies provoking perturbations during overground walking, the ability of treadmill devices to achieve highly standardised perturbation delivery and collect continuous walking patterns over long periods of time have demonstrated their utility. Split-belt treadmills seem to be more effective for provoking perturbations in comparison with single-belt treadmills. Each study attempted to create conditions that minimised participants’ awareness of the perturbation conditions and learning effects to counteract the provoked perturbations. Clear supporting evidence was observed for the delivery of perturbations to the right limb instead of the left. Future researchers are encouraged to select the perturbed limb based on side-dominance. Generally, studies instructed participants to ambulate at a self-selected velocity, although this velocity scheme does not guarantee that the perturbations delivered are equally challenging for all subjects.Bearing this in mind, authors are encouraged to compute subject-specific gait speeds to provide similar dynamic conditions among participants. Optical MoCap systems were the most used sensor systems to collect data during the experiments. Future investigations are encouraged to use EMG sensors regarding the relevant information they provide to better understand and promptly detect human responses to gait perturbations. Furthermore, the inclusion of both slip- and trip-like perturbations within the same experiment would elicit more realistic and natural LOB responses, which would promote a more effective PBT than if only a single type of perturbation was applied. The reliable application of either slip- or trip-like perturbations to mimic real-life LOB events may shorten the gap between falls in the laboratory and real-life.

Finally, future investigations should seek to standardise PBT protocols on the targeted populations to provide standard guidelines and thus encourage clinical validation of this therapeutic approach.

## Figures and Tables

**Figure 1 sensors-22-09254-f001:**
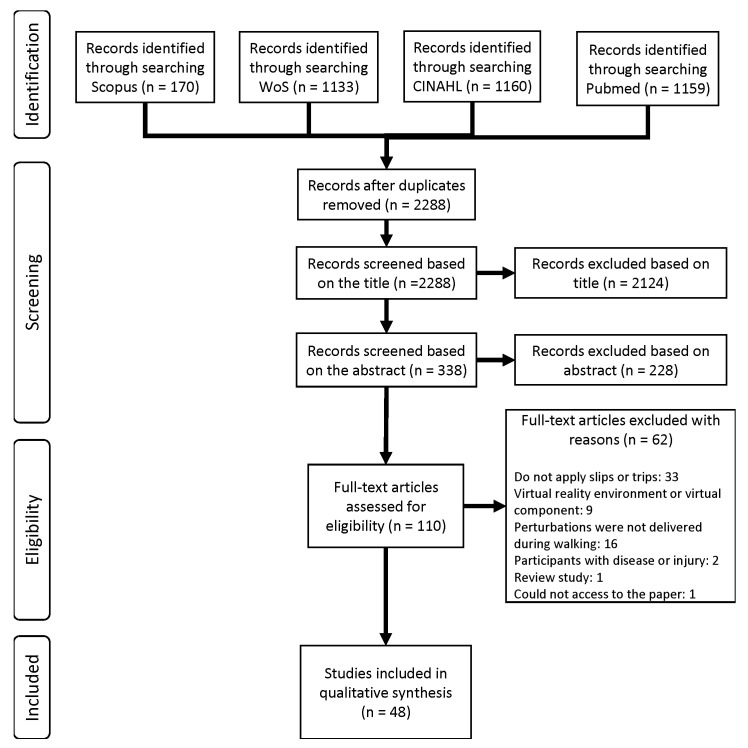
PRISMA flow diagram.

**Figure 2 sensors-22-09254-f002:**
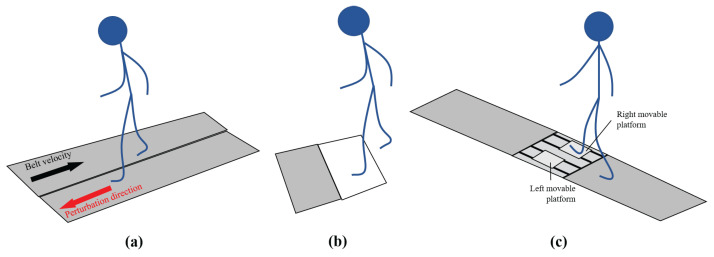
Some slip-like perturbation methods conducted in the selected studies. (**a**) Changing belt acceleration [21]. (**b**) Application of a slippery solution (gray surface) [54]. (**c**) Movable platforms [49].

**Figure 3 sensors-22-09254-f003:**
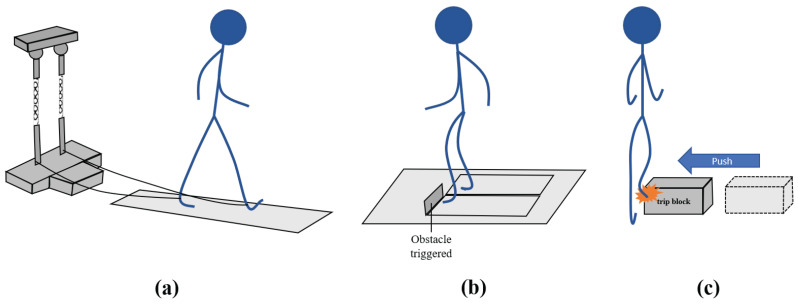
Some trip-like perturbation methods conducted in the selected studies. (**a**) Brake-and-release system [64]. (**b**) automatic obstacle trigger [68]. (**c**) Manual obstacle placement [69].

**Table 1 sensors-22-09254-t001:** Characteristics of perturbations applied in the group of 48 selected articles, where: * = some studies conducted both slips and trips, ** = some studies conducted both perturbation conditions.

Type of Perturbation (No. of Studies)	Perturbation Condition (No. of Studies)	Perturbation Mechanism	No. of Studies	Articles
Slip (40) (*)(**)	Treadmill walking (18)	Changing belt acceleration	18	[9,10,11,21,29,30,31,32,33,34,35,36,37,38,39,40,41,42]
	Overground walking (29)	Movable platforms	19	[9,10,11,12,22,30,31,33,34,43,44,45,46,47,48,49,50,51,52]
		Slippery solutions	8	[53,54,55,56,57,58,59,60]
		Novel Robotic Devices	2	[61,62]
Trip (15) (*)	Treadmill walking (6)	Changing belt acceleration	3	[41,42,63]
		Brake-and- release systems	2	[64,65]
		Tripping device	1	[66]
	Overground Walking (9)	Obstacle trigger	6	[50,51,52,60,67,68]
		Manual obstacle placement	2	[69,70]
		Novel robotic devices	1	[62]

**Table 2 sensors-22-09254-t002:** Overview of the 18 studies that performed treadmill slip-like perturbations, where: Y = young subjects, O = older subjects, MoCap = Motion Capture system, EMG = electromyography sensors, fMRI = functional Magnetic Ressonance Imaging, N\A = not available.

Study	Participants (Number/Age)	Perturbation Method	Gait Event	LOB Direction	Speed (m/s)	Perturbed Leg	Sensor Systems
Wang et al. [9]	(146/65≤)	Changing belt acceleration	heel strike	backward	Self-selected speed from 4 speed options (1.2, 1.0, 0.8 or 0.6)	Right leg	Optical MoCap; Force plate
Yang et al. [10]	(43/N\A (Y))	Changing belt acceleration	foot touchdown	backward	1.2	Right leg	Optical MoCap
Liu et al. [11]	(152/65≤)	Changing belt acceleration	N\A	backward	Self-selected speed from 4 speed options (1.2, 1.0, 0.8 or 0.6)	Right leg	Optical MoCap; Force plate
Swart et al. [21]	(30/21.6 ± 2.2)	Changing belt acceleration	heel strike	backward	1.0	Right leg	Force plate
Aprigliano et al. [29]	(10 Y/24.4 ± 2.5); (10 O/66.3 ± 5.1)	Changing belt acceleration	heel strike	backward	Normalised speed calculated for each subject	Both legs	Optical MoCap; Force plate
Lee et al. [30]	(45/74.5 ± 6.9)	Changing belt acceleration	N\A	backward	Self-selected speed from 4 speed options (1.2, 1.0, 0.8 or 0.6)	Right leg	Optical MoCap; Force plate;
Lee et al. [31]	(45/74.5 ± 6.9)	Changing belt acceleration	N\A	backward	Self-selected speed from 4 speed options (1.2, 1.0, 0.8 or 0.6)	Right leg	Optical MoCap; Force plate;
Patel et al. [32]	(10/27 ± 4)	Changing belt acceleration	N\A	backward	Self-selected	N\A	Optical MoCap; fMRI
Lee et al. [33]	(36/26.74 ± 4.9)	Changing belt acceleration	N\A	backward	1.2	Right leg	Optical MoCap; Force plate;
Liu et al. [34]	(36/N\A (Y))	Changing belt acceleration	beginning of the single stance phase	backward	1.2	Right leg	Optical MoCap; Force plate
Ding et al. [35]	(36/71.3 ± 4.7)	Changing belt acceleration	foot touchdown	backward	Self-selected	N\A	Optical MoCap
Bhatt et al. [36]	(10/26.90 ± 4.25)	Changing belt acceleration	N\A	backward	Self-selected	N\A	fMRI
Debelle et al. [37]	(17/25.2 ± 3.7)	Changing belt acceleration	heel strike	forward	1.2	Right leg	Optical MoCap; Force plate
Hirata et al. [38]	(10/21.0 ± 1.0)	Changing belt acceleration	heel strike	backward	Matched with the beat of a metronome (slow (0.9) and fast (1.6) conditions)	Both legs	Optical MoCap; Force plate
Martelli et al. [39]	(8 Y/24 ± 2.7); (8 O/65 ± 4.8)	Changing belt acceleration	heel strike	backward	Normalised speed calculated for each subject	Both legs	Optical MoCap; Force plate
Wang et al. [40]	(25/70.2 ± 5.9)	Changing belt acceleration	double-stance or single-stance phases	backward	Self-selected speed from 4 speed options (1.2, 1.0, 0.8 or 0.6)	N\A	Optical MoCap
Lee et al. [41]	(20/23.3 ± 3.3)	Changing belt acceleration	initial double- limb support	backward	Self-selected	Non-dominant leg	Optical MoCap; Force plate; EMG
Mueller et al. [42]	(13/28 ± 3)	Changing belt acceleration	heel strike	backward	1	Both legs	EMG; Plantar Pressure Insole

**Table 3 sensors-22-09254-t003:** Overview of the 29 studies that performed overground slip-like perturbations, where: Y = young subjects, O = older subjects, MA = middle-aged subjects, N\A = not available, MoCap = Motion Capture system, EMG = electromyography sensors, fMRI = functional Magnetic Resonance Imaging, * = The impulses were elicited by a robotic device that followed the subject’s motion.

Study	Participants (Number/Age)	Perturbation Method	Gait Event	LOB Direction	Speed (m/s)	Perturbed Leg	Sensor Systems
Wang et al. [9]	(146/65≤)	Movable platform	heel strike	backward	Self-selected	Right leg	Optical MoCap; Force plate
Yang et al. [10]	(43/N\A (Y))	Movable platform	step contact	backward	Self-selected	Right leg	Optical MoCap
Liu et al. [11]	(152/65≤)	Movable platfom	step contact	backward	Self-selected	Right leg	Optical MoCap; Force plate
Liu et al. [12]	(75/65≤)	Movable platform	step contact	backward	Self-selected	Right leg	Optical MoCap; Force plate
Sawers et al. [22]	(25/N\A (O))	Movable platform	heel strike	backward	N\A	Right leg	Optical MoCap; Force plate; EMG
Lee et al. [30]	(45/74.5 ± 6.9)	Movable platform	heel strike	backward	N\A	Right leg	Optical MoCap; Force plate;
Lee et al. [31]	(45/74.5 ± 6.9)	Movable platform	heel strike	backward	N\A	Right leg	Optical MoCap; Force plate;
Lee et al. [33]	(36/26.74 ± 4.9)	Movable platform	heel strike	backward	Self-selected	Right leg	Optical MoCap; Force plate;
Liu et al. [34]	(36/N\A (Y))	Movable platform	heel strike	backward	N\A	Right leg	Optical MoCap; Force plate
Wang et al. [43]	(195/72.3 ± 5.3)	Movable platform	foot contact	backward	N\A	Right leg	Optical MoCap; Force plate;
Wang et al. [44]	(195/72.3 ± 5.3)	Movable platform	foot contact	backward	N\A	Right leg	Optical MoCap; Force plate
Sawers et al. [45]	(28/N\A (O))	Movable platform	heel strike	backward	Self-selected	Right leg	Optical MoCap; Force plate; EMG
Inkol et al. [46]	(11/21.9 ± 0.3)	Movable platform	heel strike	forward, right and left	Self-selected	Dominant leg	Optical MoCap; Force plate
Liu et al. [47]	(131/71.8 ± 5.2)	Movable platform	step contact	backward	Self-selected	N\A	Optical MoCap; Force plate
Wang et al. [48]	(114/72.5 ± 5.3)	Movable platform	foot contact	backward	Self-selected	Right leg	Optical MoCap; Force plate
Wang et al. [49]	(67/72.2 ± 5.3)	Movable platform	heel strike	backward	Self-selected	Right leg	Optical MoCap; Force plate
Okubo et al. [50]	(44/65–90)	Movable platform	foot contact	backward	Matched with the beat of a metronome	Both legs	Optical MoCap
Okubo et al. [51]	(10/29.1 ± 5.6)	Movable platform	foot contact	backward	Matched with the beat of a metronome	Both legs	Optical MoCap
Okubo et al. [52]	(10 Y/20–40); (10 O/65–90)	Movable platform	foot contact	backward	Matched with the beat of a metronome	Both legs	Optical MoCap
Merril et al. [53]	(16 Y/20–31); (17 O/50–65)	Slippery solution	heel strike	unconstrained	Self-selected	Left leg	Optical MoCap; Force plate
Nazifi et al. [54]	(20/23.6 ± 2.52)	Slippery solution	heel strike	unconstrained	Self-selected	Left leg	Optical MoCap; Force plate; EMG
Ziaei et al. [55]	(22/24.5 ± 3.43)	Slippery solution	N\A	unconstrained	Self-selected	Right leg	Optical MoCap; Force plate
Soangra et al. [56]	(7 Y/22.64 ± 2.56); (7 O/71.14 ± 6.51)	Slippery solution	heel strike	unconstrained	Self-selected	Dominant leg	Optical MoCap; Force plate; EMG; Inertial sensors
O’Connel et al. [57]	(24 Y/23.75 ± 2.83); (24 MA/57.13 ± 2.83)	Slippery solution	heel strike	unconstrained	Self-selected	Left leg	Optical MoCap; Force plate; EMG
Allin et al. [58]	(108/18–66)	Slippery solution	heel strike	unconstrained	Self-selected (slightly hurried)	Dominant leg	Optical MoCap; Force plate;
Kim et al. [59]	(8/19–27)	Slippery solution	heel strike	unconstrained	N\A	Dominant leg	Optical MoCap; Force plate
Arena et al. [60]	(12/20.9 ± 2.2)	Slippery solution	heel strike	unconstrained	Between 1.45 and 1.60	Right leg	Optical MoCap; Inertial Sensors
Rasmussen et al. [61]	(6/23 ± 2.4)	Slippery solution (robotic device)	heel strike, mid-stance and toe-off	unconstrained	Self-selected	Dominant leg	Optical MoCap
Er et al. [62]	(7/25 ± 0.94)	Motor impulse (robotic device)(*)	early stance phase	backward	Self-selected	Left leg	Optical Mocap; Inertial Sensors

**Table 4 sensors-22-09254-t004:** Overview of the 6 studies that performed treadmill trip-like perturbations, where: Y = young subjects, O = older subjects, MA = middle-aged subjects, N\A = not available, MoCap = Motion Capture system, EMG = electromyography sensors.

Study	Participants (Number/Age)	Perturbations Method	Gait Event	LOB Direction	Speed (m/s)	Perturbed Leg	Sensor Systems
Lee et al. [41]	(20/23.3 ± 3.3)	Changing belt acceleration	initial double- limb support	forward	Self-selected	Non-dominant leg	Optical MoCap; Force plate; EMG
Mueller et al. [42]	(13/28 ± 3)	Changing belt acceleration	heel strike	forward	1	Both legs	EMG; Plantar Pressure Insole
Lee et al. [63]	(10/26.3 ± 4.8)	Changing belt acceleration	initial double- limb support	forward	Self-selected	Non-dominant leg	Optical MoCap, Force plate
Aprigliano et al. [64]	(8/25.9 ± 2.8)	Brake-and-release system	Swing phase	forward	1	Right leg	Optical MoCap; Inertial sensors
König et al. [65]	(24/41–62)	Brake-and-release system	Swing phase	forward	1.4	Right leg	Optical MoCap; Inertial sensors
Silver et al. [66]	(7 Y/24 ± 3.3); (4 MA/46 ± 3.0); (3 O/63 ± 3.8)	Tripping device	Early swing phase	forward	Self-selected	Left leg	Optical MoCap

**Table 5 sensors-22-09254-t005:** Overview of the 9 studies that performed overground trip-like perturbations, where: Y = young subjects, O = older subjects, N\A = not available, MoCap = Motion Capture system, EMG = electromyography sensors, * = Impulses were elicited by a robotic device that followed subject’s motion.

Study	Participants (Number/Age)	Perturbations Method	Gait Event	LOB Direction	Speed (m/s)	Perturbed Leg	Sensor Systems
Okubo et al. [50]	(44/65–90)	Obstacle trigger	Mid-swing	forward	Matched with the beat of a metronome	Both legs	Optical MoCap
Okubo et al. [51]	(10/29.1 ± 5.6)	Obstacle trigger	Mid-swing	forward	Matched with the beat of a metronome	Both legs	Optical MoCap
Okubo et al. [52]	(10 Y/20–40); (10 O/65–90)	Obstacle trigger	Mid-swing	forward	Matched with the beat of a metronome	Both legs	Optical MoCap
Arena et al. [60]	(12/20.9 ± 2.2)	Obstacle trigger	Mid-to-late swing phase	forward	Between 1.45 and 1.60	Right leg	Optical MoCap; Inertial Sensors
Er et al. [62]	(7/25 ± 0.94)	Braking impulse from a novel robotic device (*)	Terminal Swing and Mid-swing	forward	Self-selected	Left leg	Optical Mocap; Inertial Sensors
Potocanac et al. [67]	(7/24.6 ± 3.2)	Obstacle trigger	Mid-swing	forward	Self-selected	Right leg	Optical MoCap; Force plate; EMG
Wang et al. [68]	(40/67.9 ± 5.5)	Obstacle trigger	Mid-to-late swing phase	forward	N\A	Left leg	Optical MoCap; Force plate
Ko et al. [69]	(6/21.83 ± 0.75)	Manual Obstacle Placement	Swing phase	forward	N\A	Right leg	Optical Mocap
Schulz [70]	(14 Y/20–35); (25 O/66–89)	Manual Obstacle Placement	Swing Phase	forward	3 speeds: slower than preferred; preferred; and as fast as safely possible	N\A	Optical Mocap

## Data Availability

Not applicable.

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
