# Peer review of "Provoking Artificial Slips and Trips towards Perturbation-Based Balance Training: A Narrative Review"

_sensors, 2022, doi:10.3390/s22239254_

Round 1
Reviewer 1 Report
The manuscript is well written. The results are well presented and add to field of interest where a lot of work is currently under way.
I have 2 questions: why did you perform a narrative review? Your approach resembles a systematic review. With more than 40 studies it is beyond a narrative approach.
Sometimes it occurs that other groups to similar work in parallel.
This year McCrum, Okubo, Bhatt and others have published an articles that overlaps. This article is not mentioned. You must include this in the discussion and you should comment how your findings align or are different with them.
Pertubation can be produced in many ways. Slips and trips are one part. Based on the work of Maki and others many authors consider medio-lateral pertubations as a major way to intervene. Why did you focus on slips/trips only?
Finally, on Sept 30th, 2022 the first global guidelines were published. These should be mentioned in the discussion. They discuss the role of pertubation training.
Author Response
Comment 1:
Why did you perform a narrative review? Your approach resembles a systematic review. With more than 40 studies it is beyond a narrative approach
Answer to comment 1:
Thank you for this comment and for calling our attention to this detail. Although we followed PRISMA guidelines to select the articles for the review’s literature analysis, we consider this manuscript as a narrative review. It is explained since our work conducts a descriptive approach on the methods used to provoke slip- and trip-like perturbations to identify experimental aspects for future related research, namely PBT interventions, rather than a ‘quantitative’ analysis of results.
Comment 2:
This year McCrum, Okubo, Bhatt and others have published an articles that overlaps. This article is not mentioned. You must include this in the discussion and you should comment how your findings align or are different with them.
Answer to comment 2:
Thank you for the suggestion and for calling our attention for this detail. It helps to update and enrich the manuscript content.
The mentioned review analyses and discusses the key concepts regarding PBT towards the clinical implementation of this therapy. Since this is in line with the research performed in our narrative review, we updated our Introduction section including the description of this review and outlined the research questions addressed by us that were either not analysed by that review or were included in that review’s discussion. The Introduction section was updated with the 2 following parcels of text as follows:
Lines 75-79: «Furthermore, in a recent review study conducted by McCrum et al. [26], key concepts and common concerns related to PBT were discussed towards the implementation of this therapy in clinical settings. Despite the authors synthetically portrayed the current evidences on the efficacy and feasibility of PBT, a deep analysis on the current systems and methodologies used to apply slips and trips is not performed. »
Lines 99-103: «Despite review [26] analysed the second question, our review provides a deeper analysis on the comparison between perturbation delivery during treadmill and overground walking. Furthermore, although McCrum et al. [26] focused their analysis mainly on the ninth question, we attempted to perform a more concise overview about the current stage of the PBT implementation on clinical settings. »
Given the importance of discussing the current state of the feasibility of PBT implementation in the clinical settings, we also included a new research question, whose discussion was included in the Discussion section. This was the ninth question and was entitled «What is the current feasibility of PBT implementation in clinical settings? ». The objective of this research question was to portray the current evidence observed in balance and stability improvements in targeted population (older subjects) obtained by PBT sessions. Then, current barriers to implement the PBT in clinical settings are analysed, namely the lack of standardization of PBT intervention protocols, safety requirements, participants’ potential anxiety issues, equipment costs and personnel required to perform PBT. The Discussion section was updated as follows:
Lines 831-894: «4.9. What is the current feasibility of PBT implementation in clinical settings?
The emerging PBT paradigm towards fall prevention has shown encouraging results in fall rate reduction in recent years. Some experimental studies included in this review [12,40,68] depicted that PBT sessions allowed older adults to develop short- and long-term retention of fall-resisting skills, such as improving their feedforward and feedback adjustments of COM stability and body kinematics to unexpected perturbations. These findings are in line with the review conducted by McCrum et al. [26]. Liu et al.
[12] observed that older subjects fell significantly less due to slips provoked after receiving PBT in the follow-up periods of 6, 9, and 12 months compared to the single slip applied during the perturbation training. Also, studies [11,50] verified that participants which enrolled in a PBT session experienced fewer falls and had improved postural stability for perturbation recovery than participants from the control group. Okubo et al.
[52] verified that after the perturbation training completion, older adults were able to improve their margin of stability for balance recovery, which reduced their slip-induced falls from 28.6% to 14.3%. Furthermore, Wang et al. [68] reported that despite 48% of the older adults included in their study fell on the first trip, no participant fell on the trip provoked after the perturbation training.
Despite these promising results, there is still a lack of standardisation regarding the PBT protocols conducted in the scientific literature. The number of perturbations within each PBT session ranges from a single perturbation [12] to 40 perturbations [9,11]. Sessions are usually completed within one [11,52] or two days [50]. In addition, the retention of fall-resisting skills is either analysed immediately after the PBT session [11,40,50,52,68] (after 30 min [68] or 1 hour [50]), or in a long-term follow-up period
of 6 [11,12,31], 9 months [12] and 12 months [12]. This lack of standard PBT protocols hinders the implementation of PBT in clinical practice [26]. As portrayed by McCrum et al. [26], standard perturbation doses must be established. Accordingly, further research must clarify the optimal number of PBT sessions, the number of perturbations delivered within each session as well as the perturbation intensity profile during each session in order to maximise the long-term retention of the fall-resisting skills acquired during the training. In this regard, McCrum et al. [26] acknowledge the need to perform dose- response studies to uncover these optimal training conditions, which should be preferably
conducted using the more clinically appropriate treadmill-based protocols. Currently, the perturbation dosage which yields the maximum efficacy remains unknown [26]. Lee et al. [31] observed that from a treadmill PBT protocol including a lower (24 slips) and a higher (40 slips) perturbation dosage groups, only the higher dosage group showed a significantly better stability performance than the control group in the 6-month follow- up overground slip perturbation. However, Lee et al. [30] findings suggest that a higher practice dosage (40 slips) did not necessarily benefit the performance of older subjects to counteract an overground slip after a treadmill PBT session in comparison with a lower practice dosage (24 slips). While Lee et al. [31] concluded that the higher perturbation dosage might be critical for the long-term retention of fall-resisting skills, Liu et al. [12] findings suggest that a PBT session containing a single slip may be effective to prevent future falls. As such, future research must address this issue to define a standard perturbation dosage to include in PBT sessions.
Furthermore, PBT requires additional safety measures compared to conventional balance training [26]. Safety harness systems are frequently used to prevent falls during PBT sessions. These systems guarantee the subject’s safety by arresting any irreversible LOB that would lead to a fall while assuring subject’s freedom of movement during regular gait and allowing the therapist to be focused on the perturbation delivery. This equipment is usually mounted on the ceiling with a low-friction trolley-and-beam system either above the treadmill [9,11] or the overground walkway [9,11,12,52,68]. A safety harness system is required to be comfortable and well-fitted to not interfere with participants’ biomechanical responses to the perturbations and prevent their discomfort during and after the training [26,50]. In addition, it is mandatory to objectively assess the anxiety levels exhibited by the participants during the PBT sessions, as the increase of the perturbation intensity and unpredictability may cause the participant’s withdrawal from the training. Accordingly, Okubo et al. [52] pointed out the need for further research to investigate and develop multi-day PBT interventions that are suited to individuals’ ability to counteract the perturbations and anxiety levels. It is also noteworthy that the costs related to the currently used perturbation systems as well as the expert personnel required to operate them may hinder the implementation of PBT in clinical settings [3].
Thereby, new experimental studies should attempt to fill the current gaps to provide guidance to clinicians and develop standard clinical practice guidelines for PBT implementation [26]. »
Comment 3:
Perturbation can be produced in many ways. Slips and trips are one part. Based on the work of Maki and others many authors consider medio-lateral perturbations as a major way to intervene. Why did you focus on slips/trips only?
Answer to comment 3:
Thank you for your question, as it helps to clarify the main objectives of this narrative review as well as future investigation directions on the research topics analysed.
As pointed out in previous papers (references [5] and [19] from our manuscript), slips and trips are the most prevalent events which precede falls. As such, these perturbation events are the most commonly applied during experimental protocols and therefore were considered the most relevant to address in PBT interventions. Although medio-lateral perturbations are also very important to include in a broader analysis of PBT, perturbations in the antero-posterior direction of motion are considered more common and receive more attention in scientific literature studies (reference [74] from our manuscript). Based on the literature findings, we considered only the papers which applied either slip- or trip-like perturbations, leading to 48 reviewed studies. However, we think that future investigations should address medio-lateral perturbations to portray the mechanisms used in the literature to induce these perturbations as well as to uncover their role in PBT interventions. Once again, we would like to thank you for your comment.
Comment 4:
Finally, on Sept 30th, 2022 the first global guidelines were published. These should be mentioned in the discussion. They discuss the role of pertubation training.
Answer to comment 4:
Thank you for your comment. We were sorry but we could not find the global guidelines you mentioned regarding perturbation training. Can you please provide more specific information about these guidelines? Thank you for your time in advance.

Reviewer 2 Report
The article is a review paper on the main gait-perturbation methods reported in the literature for treadmill and overground walking. It focuses more specifically on slip-like and trip-like perturbations, and the authors discuss and ‘compare’ different factors and parameters, such as treadmill vs overground, the type of method used to induce the perturbation, the gait-phase during which the perturbation is applied, gait speed, perturbed leg, as well as the equipment / sensors used for the experiments.
The paper is well structured, well written, and therefore easy to read. It is very ‘descriptive’ (i.e., no real ‘quantitative’ analysis of effects or results, such as that often found in meta-analysis, for instance including forest plots), but I guess it is okay considering the aim of the paper, which is defined by the authors as a ‘narrative review’.
My main concern relates to the fact that the training aspect is almost not addressed, although ‘perturbation-based balance training’ is mentioned in the title of the paper. Put differently, the authors describe and discuss different factors, but they do not state to which extent those ACTUALLY affect training, and more specifically training-evoked improvements. To me, it’s a bit like describing the different therapies and medications used to treat an ailment, without knowing what works best. While I believe the paper is informative as it currently is, as it presents the main advantages and disadvantages of the different methods and parameters, I think it would be even more informative with training-related information, if this information is ‘available’.
On a side note, though sensors-related aspects are addressed, these do not constitute the ‘core’ of the paper. I’m only mentioning this point because sensors-related aspects do constitute the core aims and scope of the journal.
Author Response
Comment 1:
The paper is well structured, well written, and therefore easy to read. It is very ‘descriptive’ (i.e., no real ‘quantitative’ analysis of effects or results, such as that often found in meta-analysis, for instance including forest plots), but I guess it is okay considering the aim of the paper, which is defined by the authors as a ‘narrative review’.
Answer to comment 1:
Thank you for your comment. Since the main objective of this manuscript was to present the different methods used in the scientific literature to provoke slip- and trip-like perturbations and identify key aspects to consider in future research, this paper was mainly intended to be descriptive. However, as pointed out in the next comment (Comment 2.2), we clarified the perturbation training effect and training-evoked improvements of balance on targeted patients in order to provide a quantitative analysis of the improvements induced by PBT towards the targeted subjects’ fall risk decrease.
Comment 2:
My main concern relates to the fact that the training aspect is almost not addressed, although ‘perturbation-based balance training’ is mentioned in the title of the paper. Put differently, the authors describe and discuss different factors, but they do not state to which extent those ACTUALLY affect training, and more specifically training-evoked improvements. To me, it’s a bit like describing the different therapies and medications used to treat an ailment, without knowing what works best. While I believe the paper is informative as it currently is, as it presents the main advantages and disadvantages of the different methods and parameters, I think it would be even more informative with training-related information, if this information is ‘available’.
Answer to comment 2:
Thank you for this comment and for calling our attention to this important research topic. Although the main objective of this manuscript was to review the different methodologies used to provoke both slip- and trip-like perturbations and identify key concepts to consider in future research, namely PBT interventions, we need to clarify the perturbation training effect and training-evoked improvements of balance on targeted patients. Currently, there is not an optimal PBT intervention protocol to maximize training-evoked balance and stability improvements. Different articles conclude opposite findings, namely recommending higher or lower perturbation dosage, i.e., number of perturbations, delivered to the participants.
Given the importance of discussing the current state of the feasibility of PBT implementation in the clinical settings, we included a new research question that address the currently reported training effects of PBT and complemented it with requirements needed to fulfil to further conduct the clinical validation of PBT interventions. This was
the ninth question and was entitled «What is the current feasibility of PBT implementation in clinical settings? ». The objective of this research question was to portray the current evidence observed in balance and stability improvements in targeted population (older subjects) obtained by PBT sessions. Then, current barriers to implement the PBT in clinical settings are analysed, namely the lack of standardization of PBT intervention protocols, safety requirements, participants’ potential anxiety issues, equipment costs and personnel required to perform PBT. The Discussion section was updated as follows:
Lines 831-894: «4.9. What is the current feasibility of PBT implementation in clinical settings?
The emerging PBT paradigm towards fall prevention has shown encouraging results in fall rate reduction in recent years. Some experimental studies included in this review [12,40,68] depicted that PBT sessions allowed older adults to develop short- and long-term retention of fall-resisting skills, such as improving their feedforward and feedback adjustments of COM stability and body kinematics to unexpected perturbations. These findings are in line with the review conducted by McCrum et al. [26]. Liu et al.
[12] observed that older subjects fell significantly less due to slips provoked after receiving PBT in the follow-up periods of 6, 9, and 12 months compared to the single slip applied during the perturbation training. Also, studies [11,50] verified that participants which enrolled in a PBT session experienced fewer falls and had improved postural stability for perturbation recovery than participants from the control group. Okubo et al.
[52] verified that after the perturbation training completion, older adults were able to improve their margin of stability for balance recovery, which reduced their slip-induced falls from 28.6% to 14.3%. Furthermore, Wang et al. [68] reported that despite 48% of the older adults included in their study fell on the first trip, no participant fell on the trip provoked after the perturbation training.
Despite these promising results, there is still a lack of standardisation regarding the PBT protocols conducted in the scientific literature. The number of perturbations within each PBT session ranges from a single perturbation [12] to 40 perturbations [9,11]. Sessions are usually completed within one [11,52] or two days [50]. In addition, the retention of fall-resisting skills is either analysed immediately after the PBT session [11,40,50,52,68] (after 30 min [68] or 1 hour [50]), or in a long-term follow-up period
of 6 [11,12,31], 9 months [12] and 12 months [12]. This lack of standard PBT protocols hinders the implementation of PBT in clinical practice [26]. As portrayed by McCrum et al. [26], standard perturbation doses must be established. Accordingly, further research must clarify the optimal number of PBT sessions, the number of perturbations delivered within each session as well as the perturbation intensity profile during each session in order to maximise the long-term retention of the fall-resisting skills acquired during the training. In this regard, McCrum et al. [26] acknowledge the need to perform dose- response studies to uncover these optimal training conditions, which should be preferably conducted using the more clinically appropriate treadmill-based protocols. Currently, the perturbation dosage which yields the maximum efficacy remains unknown [26]. Lee et al. [31] observed that from a treadmill PBT protocol including a lower (24 slips) and
a higher (40 slips) perturbation dosage groups, only the higher dosage group showed a significantly better stability performance than the control group in the 6-month follow- up overground slip perturbation. However, Lee et al. [30] findings suggest that a higher practice dosage (40 slips) did not necessarily benefit the performance of older subjects to counteract an overground slip after a treadmill PBT session in comparison with a lower practice dosage (24 slips). While Lee et al. [31] concluded that the higher perturbation dosage might be critical for the long-term retention of fall-resisting skills, Liu et al. [12] findings suggest that a PBT session containing a single slip may be effective to prevent future falls. As such, future research must address this issue to define a standard perturbation dosage to include in PBT sessions.
Furthermore, PBT requires additional safety measures compared to conventional balance training [26]. Safety harness systems are frequently used to prevent falls during PBT sessions. These systems guarantee the subject’s safety by arresting any irreversible LOB that would lead to a fall while assuring subject’s freedom of movement during regular gait and allowing the therapist to be focused on the perturbation delivery. This equipment is usually mounted on the ceiling with a low-friction trolley-and-beam system either above the treadmill [9,11] or the overground walkway [9,11,12,52,68]. A safety harness system is required to be comfortable and well-fitted to not interfere with participants’ biomechanical responses to the perturbations and prevent their discomfort during and after the training [26,50]. In addition, it is mandatory to objectively assess the anxiety levels exhibited by the participants during the PBT sessions, as the increase of the perturbation intensity and unpredictability may cause the participant’s withdrawal from the training. Accordingly, Okubo et al. [52] pointed out the need for further research to investigate and develop multi-day PBT interventions that are suited to individuals’ ability to counteract the perturbations and anxiety levels. It is also noteworthy that the costs related to the currently used perturbation systems as well as the expert personnel required to operate them may hinder the implementation of PBT in clinical settings [3].
Thereby, new experimental studies should attempt to fill the current gaps to provide guidance to clinicians and develop standard clinical practice guidelines for PBT implementation [26]. »
Comment 3:
On a side note, though sensors-related aspects are addressed, these do not constitute the ‘core’ of the paper. I’m only mentioning this point because sensors-related aspects do constitute the core aims and scope of the journal.
Answer to comment 3:
Thank you for your comment and for calling our attention to this detail. Although sensors- related aspects are not the ‘core’ of our manuscript, sensor systems are portrayed as fundamental to study, develop, and perform PBT interventions. Sensor data is crucial to assess the training effects of PBT concerning the analysis of balance and stability improvements of targeted patients. In other words, sensors have a crucial role in the study of the effectiveness of the PBT interventions.

Round 2
Reviewer 1 Report
all points addressed
Reviewer 2 Report
The article remains very 'descriptive' (i.e., no real quantification of effects) but the authors took the comments into account and clarified their 'stance / approach', notably by adding a long paragraph in the discussion. I'm satisfied with the revised version of the manuscript